# Diversity of Marek’s Disease Virus Strains in Infections in Backyard and Ornamental Birds

**DOI:** 10.3390/ani14192867

**Published:** 2024-10-05

**Authors:** Ruy D. Chacón, Christian J. Sánchez-Llatas, Claudete S. Astolfi-Ferreira, Tânia Freitas Raso, Antonio J. Piantino Ferreira

**Affiliations:** 1Department of Pathology, School of Veterinary Medicine, University of São Paulo, Av. Prof. Orlando Marques de Paiva, 87, São Paulo 05508-900, Brazil; ruychaconv@alumni.usp.br (R.D.C.); csastolfi@gmail.com (C.S.A.-F.); tfraso@usp.br (T.F.R.); 2Department of Genetics, Physiology, and Microbiology, Faculty of Biology, Complutense University of Madrid (UCM), 28040 Madrid, Spain; chsanc01@ucm.es

**Keywords:** *Mardivirus gallidalpha2*, oncogenic virus, *meq*, backyard chicken, Silkie chicken, Indian Giant, red junglefowl, Indian peafowl, phylogenetic tree, selective pressure

## Abstract

**Simple Summary:**

Marek’s disease virus is a pathogen capable of inducing tumors, immunosuppression, and paralysis in birds. The degree of virulence is associated with oncogenes such as the *meq* gene, which is constantly subjected to mutations that can exacerbate clinical manifestations and aid immune escape. This study reports the detection of this virus in backyard and ornamental birds from Brazil and Peru. Through molecular and evolutionary analyses, we determined the circulation of various strains with distinct genetic virulence profiles. Additionally, we identified potential sources of introduction and distribution of this virus among the studied birds, as well as a link between backyard and commercial birds. Given that different bird species exhibit varying degrees of susceptibility or resistance to this virus, the detection and characterization of these strains are crucial for monitoring and preventing potential epidemiological outbreaks in other species.

**Abstract:**

Marek’s disease is caused by *Mardivirus gallidalpha2*, commonly known as Marek’s disease virus (MDV). This pathogen infects various bird species resulting in a range of clinical manifestations. The *meq* gene, which is crucial for oncogenesis, has been extensively studied, but molecular investigations of MDV in noncommercial South American birds are limited. This study detected MDV in backyard and ornamental birds from Brazil and Peru and characterized the *meq* gene. MDV was confirmed in all seven outbreaks examined. Three isoforms of *meq* (S-*meq*, *meq*, and L-*meq*) and two to seven proline repeat regions (PRRs) were detected among the sequenced strains. At the amino acid level, genetic profiles with low and high virulence potential were identified. Phylogenetic analysis grouped the sequences into three distinct clusters. Selection pressure analysis revealed 18 and 15 codons under positive and negative selection, respectively. The results demonstrate significant MDV diversity in the studied birds, with both high and low virulence potentials. This study highlights the importance of monitoring and characterizing circulating MDV in backyard and ornamental birds, as they can act as reservoirs for future epidemiological outbreaks.

## 1. Introduction

Marek’s disease (MD) is a lymphoproliferative disease of birds that is characterized by the rapid development of lymphoid tumors, immunosuppression, and paralysis. It occurs in four main clinical forms: visceral, neural, cutaneous, and ocular [1,2]. The causative agent is Marek’s disease virus (MDV-1), an alphaherpesvirus classified as *Mardivirus gallidalpha2* [3]. Other *Mardivirus* serotypes related to MD include *Mardivirus gallidalpha3* (MDV-2), which is considered nonpathogenic, and *Mardivirus meleagridalpha1* (HVT or MDV-3), which exclusively affects turkeys [1,3]. Viral particles consist of hexagonal nucleocapsids (85–100 nm) enveloped by an irregular, amorphous structure (up to 400 nm) [1,4]. Since its initial descriptions were recorded, MD has evolved, transitioning from neural to tumor manifestations. This shift has increased its virulence and mortality rate [5]. MDVs are categorized into pathotypes based on virulence: mild (mMDV), virulent (vMDV), very virulent (vvMDV), and very virulent plus (vv+MDV) strains [6].

Vaccination has been a primary strategy for MD prevention, using for monovalent or bivalent strains SB-1 (MDV-2), FC126 (MDV-3), and, since the 1990s, CVI988 (MDV-1) [1,5]. While vaccination programs have advanced and are routinely implemented, MD outbreaks have continued to occur in recent years [1,7,8,9,10]. For backyard and ornamental birds, given their lower biosecurity and high economic/conservation value, vaccination is strongly recommended, but less frequently practiced and regulated [11,12,13,14,15].

The complete MDV genome has an approximate length of 178 kb and comprises two unique regions: the long unique region (UL) and the short unique region (US) [16]. These regions are flanked by identical inverted repeats of terminal and internal regions called TRL and IRL (flanking UL), and TRS and IRS (flanking US). MDVs are known to encode more than 100 functional proteins [16]. Several of these have homologs in alphaherpesviruses, while others are unique to MDVs and are important for pathogenesis and evading the host immune response [1]. Some of these unique genes include latency-associated transcripts (LATs), Meq (Marek’s EcoQ), vIL-8, viral lipase pp38/pp24, the 1.8 kb gene family, telomerase RNA (vTR), and MDV-encoded microRNAs [1]. Among these genes, the *meq* oncogene is particularly important in terms of pathogenesis and immunosuppression. This gene has a standard form that is 339 amino acids in length [16]. However, other isoforms have been described, such as long-*meq* (L-*meq*), very long-*meq* (VL-*meq*), short-*meq* (S-*meq*), and very short-*meq* (VS-*meq*) [5,12,15,17]. The presence of polymorphisms in the basic region (BR) and leucine zipper (ZIP) domains, as well as the number of proline repeats in the transactivation domain (TAD), has been associated with variations in virulence [18,19,20].

MDV infections have been reported in various bird orders, primarily in commercial species but also backyard, ornamental, and free-living birds [4,11,12,13,14,15,21,22]. While MDV infections primarily affect *Galliformes* (chickens, turkeys, quails, pheasants, and partridges), occasional cases have been documented in *Anseriformes* (ducks, geese, and swans), *Columbiformes* (pigeons), *Falconiformes* (kestrels), *Gruiformes* (cranes), *Passeriformes* (sparrows), *Strigiformes* (owls), and peafowl, as well as in the subspecies *Gallus gallus bankiva* and *lafayettii* [1,4,23,24,25].

MDV is distributed worldwide. However, in South America, molecular studies are scarce and limited to Brazil and Colombia [7,8,9,10,11,23]. For commercial birds, studies in these countries have mainly focused on layers, revealing the circulation of different strains of MDV with various levels of virulence, including vv+MDV [7,8,9,10]. In one of these studies, selective pressure events and potential routes of MDV introduction to South America in recent years were inferred [8]. Conversely, studies of ornamental and free-living birds are mainly limited to isolated cases in Brazil, where MDV was detected in co-infection with reticuloendotheliosis virus (REV) in backyard chickens [11] and in white peafowl (*Pavo cristatus*) in a zoo [23].

This study aimed to investigate the prevalence of Marek’s disease virus (MDV) in backyard and ornamental birds from Brazil and Peru. Additionally, we conducted molecular characterization and a selective pressure analysis of the *meq* oncogene in the detected MDV strains.

## 2. Materials and Methods

### 2.1. Clinical Cases and Molecular Detection of Oncogenic Viruses

The present study included a collection of cases of outbreaks involving backyard and ornamental birds affected by MD, with no history of MD vaccination in most cases. Various tissue samples were sent to the Laboratory of Avian Diseases at the School of Veterinary Medicine, University of São Paulo, Brazil (Table 1). These samples were macerated in phosphate-buffered saline (PBS) at pH 7.2 at a 1:1 ratio and subjected to three freezing and thawing cycles at −80 °C and 56 °C. Then, they were centrifuged at 12,000× *g* for 20 min at 4 °C, and 200 µL of supernatant was collected. Nucleic acid extraction was performed using the BioGene Viral kit (Quibasa - Química Básica Ltda., Belo Horizonte, Brazil). The detection of MDV and the differentiation of vaccine strain CVI988 from field strains were performed by real-time PCR (qPCR) according to previously published conditions [26], using the PowerUp™ SYBR^®^ Green Master Mix (Applied Biosystems, Austin, TX, USA) in a QuantStudio3 Real Time PCR System (Applied Biosystems, Marsiling, Singapore). To verify the presence of other oncogenic viruses, avian leukosis virus (ALV) and reticuloendotheliosis virus (REV) were detected in accordance with previous studies [27]. The first case (USP-386), previously described in [11], was further investigated in this study by sequencing the *meq* oncogene. The cases are described below.

Case 1: A 12-week-old backyard hen from a flock of 40 birds exhibited symptoms of apathy, anorexia, and facial cyanosis. Upon examination, multiple visceral tumors, polyneuritis, splenomegaly, hepatomegaly, and thymic atrophy were observed. The flock experienced a morbidity rate of 50% and a mortality rate of 10%.Case 2: A 3-month-old Indian Giant hen from a flock of 160 birds exhibited symptoms of apathy, anorexia, and paralysis. Upon examination, splenomegaly and hepatomegaly were observed. The flock experienced a mortality rate of 23%, primarily due to sudden deaths.Case 3: A 2-year-old red junglefowl rooster (*Gallus gallus bankiva*) from a flock of 100 birds exhibited symptoms of apathy, anorexia, and paralysis. Upon examination, splenomegaly, hepatomegaly, and multiple visceral tumors were observed. Each bird in this flock was individually housed. Only two roosters were affected, with morbidity and mortality rates of 2% each.Case 4: A 35-week-old backyard hen was found dead without exhibiting any obvious signs in the preceding days. Upon examination, visceral tumors were observed. The bird had been vaccinated with a dose of HVT on its first day of life. This was the only affected bird in the flock (the total number of birds in the flock was not recorded).Case 5: A 4-year-old female Indian peafowl (*Pavo cristatus*) exhibited symptoms of apathy, anorexia, paralysis, and facial cyanosis. A complete blood count suggested myelophthisis of lymphoid cells and revealed normocytic, normochromic anemia, along with a leukogram that showed leukocytosis, heterophilia, lymphocytosis, eosinophilia, and monocytosis, with 7% atypical lymphocytes, 5% large lymphocytes, and some heterophils with accentuated toxic granulation. The platelet count was normal. A radiographic examination revealed osteopenia and bone density loss, mainly in the pelvic region. Among the three Indian peafowl housed together, this was the only one that showed clinical signs. The bird died a few days after examination.Case 6: A one-year-old Silkie rooster exhibited neurological symptoms, including altered proprioception and intermittent imbalance. Among the three Silkie chickens housed together, this was the only one that showed clinical signs. The bird made a full recovery two weeks after examination.Case 7: A backyard rooster exhibited symptoms of apathy and paralysis. This was the only affected bird in the flock (the total number of birds in the flock was not recorded). The bird made a full recovery a few weeks after the examination.

Additional information is detailed in Table 1.

**Table 1 animals-14-02867-t001:** Epidemiological and clinical information of the studied cases of Marek’s disease.

Case	ID	Host	Comon Name	Year	City/Country	Sample	MD Form
1	USP-386	*Gallus gallus domesticus*	Creole Hen	2010	São Paulo/Brazil	Liver + Spleen	Visceral
2	USP-1171	*Gallus gallus domesticus*	Indian Giant	2018	São Paulo/Brazil	Liver + Spleen + Sciatic nerves	Neural
3	USP-1540	*Gallus gallus bankiva*	Red Junglefowl	2019	Lima/Peru	Spleen ^A^	Visceral
4	USP-1790	*Gallus gallus domesticus*	Creole Hen	2020	São Paulo/Brazil	Liver + Lungs	Visceral
5	USP-1873	*Pavo cristatus*	Indian Peafowl	2020	São Paulo/Brazil	Blood ^A^	Neural
6	USP-2429	*Gallus gallus domesticus*	Silkie Chicken	2022	São Paulo/Brazil	Feathers	Neural
7	USP-2583	*Gallus gallus domesticus*	Creole Hen	2022	São Paulo/Brazil	Feathers	Neural

^A^ These samples were collected in FTA^TM^ Whatman^TM^ Classic Cards (GE Healthcare, Buckinghamshire, UK).

### 2.2. Meq Gene Sequencing

To characterize the detected MDV strains, the *meq* gene was amplified with primers CCGCACACTGATTCCTAGGC and AGAAACATGGGGCATAGACG, as previously reported [9]. The PCR primers were specifically designed to target the flanking regions of the *meq* gene. This resulted in the amplification of a 1148 bp product (based on the size of the standard *meq* isoform of the reference strain Md5, accession number NC_002229), which included the complete 1020 bp *meq* gene sequence and an additional 128 bp from the upstream and downstream regions. For the L-*meq* isoform of the reference strain CVI988, accession number DQ530348.1, the expected size was 1325 bp, which included the flanking 128 bp plus the 1197 bp of the *meq* gene.

PCR amplification was performed using a reaction mix containing 0.2 mM of each dNTP, 2 mM of MgCl_2_, 0.6 mM of each primer, 0.75 U of Platinum^TM^ Taq DNA Polymerase, and 1X PCR Buffer (Thermo Fisher Scientific Baltics UAB, Vilnius, Lithuania). The thermal conditions included an initial step at 94 °C for 2 min, followed by 36 cycles at 94 °C for 1 min, 55 °C for 1 min, and 72 °C for 2 min, and a final elongation step at 72 °C for 10 min. The amplified bands were purified from a 1.5% agarose gel using an Illustra™ GFX PCR and Gel Band Purification Kit (Cytiva, Marlborough, MA, USA). Sequencing was performed using Sanger technology on a 3500xL Genetic Analyzer with the BigDye™ Terminator v3.1 Cycle Sequencing Kit (Applied Biosystems, Carlsbad, CA, USA). The electropherograms of the sense and antisense reads were assembled and analyzed using Geneious Prime^®^ 2020.2.4. This program was also used to transcribe the DNA sequences into their deduced amino acids using the standard genetic code to perform polymorphism analysis.

### 2.3. Phylogenetic Analysis

To carry out the phylogenetic analyses, a set of 381 complete *meq* sequences previously reported in [8] was analyzed along with the sequences generated in this study (Appendix A). The sequences were aligned using the MAFFT program with an iterative refinement method (FFT-NS-i) [28]. The best nucleotide evolutionary substitution model was estimated using ModelTest-NG v0.1.7 [29]. To infer the phylogenetic tree, the maximum likelihood method was applied with the RAxML program and 1000 bootstrap replicates [30]. The generated tree was processed and graphically edited using the iTOL v6 program [31].

### 2.4. Selection Pressure Analysis

For this analysis, the alignment of the complete *meq* sequences was used. Initially, positive selection at the gene level was evaluated using the BUSTED method. The FEL, FUBAR, and SLAC methods were subsequently employed to identify codons that underwent pervasive positive or negative selection. Finally, the MEME method was applied to detect codons undergoing positive episodic selection. All these methods were implemented using the Datamonkey 2.0 web platform [32].

## 3. Results

### 3.1. Molecular Detection of Oncogenic Viruses

After the molecular assays were performed, the presence of field MDV strains was confirmed in all cases (Table 2). REV was detected in one case (USP-386), while ALV was not detected in any case. 

After confirming the presence of MDV in the studied cases, amplification of the *meq* gene revealed three different sizes: 1148 bp, 1025 bp, and 1325 bp (Figure 1). Two strains (USP-386 and USP-2429) presented sizes of 1148 bp, which correspond to the standard *meq* isoform (*meq*, 1020 bp). One strain (USP-1171) presented a size of 1025 bp, which corresponds to the short isoform of *meq* (S-*meq*, 897 bp). Three strains (USP-1540, USP-1790, and USP-1873) presented sizes of 1325 bp, which correspond to the long isoform of *meq* (L-*meq*, 1197 bp). The USP-2583 strain could not be amplified by end-point PCR, possibly due to the low viral load of the sample (CT > 36). For the CVI988 vaccine strain, two bands were observed, corresponding to the standard and L-*meq* isoforms.

### 3.2. Sequence and Phylogenetic Analysis of Meq Gene

The inferred phylogenetic tree revealed the distribution of the MDV sequences into eight clusters (Figure 2). The strains from this study were distributed across three different clusters. The three strains with L-*meq* isoforms (USP-1540, USP-1790, and USP-1873) were grouped in Cluster C2. The USP-1790 strain was closely related to various virulent strains (BC-1, JM102, 02LAR, 04CRE, FT158, and MPF57) and the attenuated strain 814. The USP-1540 and USP-1873 strains were located near the moderately virulent strain CU-2. The USP-386 strain was placed in Cluster C5, near the virulent strain 571 and close to Brazilian strains detected in commercial chickens (defined as Genotype II by Chacón et al., 2024 [8]). Finally, strains USP-1171 and USP-2429 were positioned in Cluster C7, along with strains from Colombia and an S-*meq* strain from Japan.

With respect to the amino acid polymorphisms of the *meq* gene, the three strains of the L-*meq* form (USP-1540, USP-1790, and USP-1873) presented seven PRRs and shared the polymorphisms 71S, 139T, and 194delP (Table 3). Additionally, the USP-1540 and USP-1873 strains presented the 80D and 115V polymorphisms, while USP-1790 presented the 80V polymorphism and the novel 337E polymorphism. The USP-386 strain (standard *meq*) presented five PRRs and only the 139T polymorphism. Strains USP-1171 (S-*meq*) and USP-2429 (standard *meq*) presented two and three PRRs, respectively, and shared similar polymorphism profiles including 110S, 115V, 139T, 151I, 176A (which eliminates a PRR), and 180A. Additionally, USP-2429 contains the polymorphism 217A and the novel 182D. Regarding PRR sites, one was lost in USP-2429 due to the 217A polymorphism, and three PRR sites were lost in USP-1171 due to the loss of 41 amino acids compared to the standard *meq* size (339 aa–298 aa). The obtained sequences were submitted to GenBank under the accession numbers PP783759 to PP783764.

### 3.3. Selection Pressure Analysis

The first analysis was performed with BUSTED, which provided a gene-wide test (not site-specific) to detect positive selection. The result, based on the likelihood ratio test, showed evidence of episodic diversifying selection in the analyzed data (*p* = 0.0000035).

In the site-specific selective pressure analyses, 18 codons were found to be under positive selection. Among these, thirteen sites were detected to be under pervasive positive pressure (identified by the FEL, FUBAR, and SLAC methods), with nine of these sites (71, 88, 110, 151, 176, 180, 194, 296, and 418) detected by two or three methods (Table 4). Additionally, 14 sites were detected to be under episodic positive pressure (identified by the MEME method).

Fifteen sites were detected to be under pervasive negative pressure (Table 4), with seven of these sites (49, 114, 182, 208, 332, 348, and 377) identified by two or three methods.

Some polymorphisms in the MDV strains studied (Table 3) correspond to those sites under positive pressure. In the BR domain, polymorphisms were found at sites 71 and 80 (in the three strains with L-*meq*: USP-1540, USP-1790, and USP-1873). In the ZIP domain, polymorphisms were detected at site 110 (in strains USP-1171 and USP-2429). In the TAD domain, polymorphisms were found at sites 139 (in all six strains studied), 151, 176, 180, and 217 (in the USP-1171 and USP-2429 strains), with deletions at site 194 (in USP-1171, USP-1540, USP-1790, and USP-1873). On the other hand, the 182D polymorphism in the USP-2429 strain was found to be under negative pressure.

## 4. Discussion

Marek’s disease is known to affect a considerable variety of birds, primarily *Galliformes*. Over the past few decades, numerous reports have demonstrated its ubiquity in commercial birds as well as in backyard and free-living birds. This study investigated a series of MD outbreaks with molecular characterization in backyard and ornamental birds from Brazil and, for the first time, from Peru. MDV has been widely reported in backyard birds [11,12,13,15,33,34]. Other reports include the presence of MDV or antibodies in ornamental birds such as red junglefowl (*G. gallus bankiva*), Ceylon junglefowl (*Gallus lafayettii*), Yokohama chickens, Silkie chickens, and Indian peafowl [4,23,25,35,36,37,38,39]. This study provides the first report of MDV in Indian Giant roosters. Moreover, to our knowledge, this is the first time that complete *meq* sequences of strains detected in some of these birds (red junglefowl, Indian Giant roosters, and Indian peafowl) have been obtained. Molecular and evolutionary studies were conducted to characterize these strains.

The presence of oncogenic viruses has been reported in backyard, ornamental, and free-living birds [1,11,27]. In this study, in addition to MDV, reticuloendotheliosis virus (REV) was detected in only one case, that of a backyard chicken with clinical symptoms of the visceral form of MD. REV is known to be associated with immunosuppression and can aggravate the clinical presentation of MDV when coinfected [40,41]. REV circulation has been reported in Brazil in commercial and backyard chickens as well as Muscovy ducks and wild turkeys [11,42,43,44].

The *meq* oncogene is known for its length polymorphism, with up to five isoforms reported to date, including the most recent, the S-*meq* and VS-*meq* versions [12,15,17,45]. There may even be strains presenting two isoforms simultaneously, as in the case of the vaccine strain CVI988, which was also observed in this study. This phenomenon has been previously reported and is associated with the processes of passaging and attenuation [5,17,18]. In this study, we identified three isoforms, L-*meq*, standard *meq*, and S-*meq*, suggesting considerable diversity among these groups of birds. Traditionally, it has been suggested that larger *meq* sizes (and greater numbers of PRRs) are associated with lower levels of virulence and pathogenicity [19,20]. However, experimental studies have shown opposing results, indicating the potential contribution of other functions or amino acid polymorphisms of *meq* [46,47]. In this study, all cases with visceral Marek’s disease (MD) resulted in death and exhibited either the standard or L-*meq* isoform, with five and seven PRRs, respectively. In contrast, only two of the four cases with neural MD resulted in death, and three of these sequenced neural cases exhibited two, three, and seven PRRs.

Phylogenetic analysis resulted in the distribution of sequences into eight clusters, with the sequences from this study clustered into three groups. Cluster C2 shows significant geographical dispersion, containing strains from various regions across almost all continents. This cluster contains diverse isoforms including standard *meq*, S-*meq*, L-*meq*, and VL-*meq* [14,20,48]. It also includes strains isolated from *Anseriformes* and backyard chickens [12,22]. These characteristics contribute to the great diversity within this cluster. Notably, the present study adds a strain detected in red junglefowl from Peru, a strain from backyard chickens in Brazil, and a strain from Indian peafowl from Brazil to this cluster. On the other hand, this study grouped the backyard chicken strain USP-386 (from 2010) in Cluster C5. This is particularly interesting because C5 contains all the strains defined as Genotype II from Brazil that were detected in commercial birds with MD between 2018 and 2022 [8]. These findings suggest a common ancestral relationship and potential links between backyard chickens and commercial birds. Finally, Cluster C7 included strains USP-1171 and USP-2429, which were detected in an Indian giant rooster and a Silkie chicken, respectively. Cluster C7 is characterized by a small number of PRRs (between two and four). In proximity to the strains from this study, Cluster C7 also contained strains from Colombia [9], China [49], and an S-*meq* strain from Japan [50].

Some *meq* amino acid polymorphisms have been shown to significantly influence increased virulence, enhanced shedding, and immune escape from vaccines [46]. According to the distribution of polymorphisms, our three strains from C2 with the L-*meq* isoform presented profiles (71S, 77E, 119C, 153P, 176P, 217P, and 277L) associated with less virulent strains [19,46]. The profile of the USP-386 strain was very similar to that of the previous strains except for 71A. In contrast, the USP-1171 and USP-2429 strains (from C7) presented profiles associated with more virulent strains (71A, 115V, 176A, 180A, 217A, or 217delP).

Selective pressure is an evolutionary mechanism that has been reported in MDV, specifically involving the *meq* oncogene [8,51]. In this study, we identified several polymorphisms associated with greater virulence that are subject to positive pressure, confirming the findings of previous studies [8,51]. This study also identifies a series of sites subjected to negative pressure. Interestingly, the novel 182D polymorphism detected in the USP-2429 strain from a Silkie chicken is located at a negative pressure site. The roles of some of these sites in processes such as oncogenesis, pathogenicity, and immunological response have been reported [19,49,52,53,54]. The distribution of evolutionary forces along *meq* codons may suggest important functions during MDV infection, as has been observed in other viruses affecting commercial and wild birds [55,56].

Considering all the results, there is a considerable diversity of MDVs among the cases presented in this study. This diversity can be associated with various sources of origin as previously reported [8,12]. Furthermore, this diversity is reflected in virulence potential, according to the *meq* amino acid profiles found. However, no direct relationship was observed between the sequences studied and their clinical presentation. There are varying degrees of susceptibility and genetic resistance among different breeds of commercial birds and between different host species [1,4]. Other factors influencing the degree of clinical manifestation include biosecurity measures (which are less strict for backyard and ornamental birds than for commercial birds), coinfection with other pathogens, age of the birds, immune status, lack of vaccination, and environmental factors, among others [18,57]. 

## 5. Conclusions

The results of this study demonstrate significant MDV diversity in backyard and ornamental birds, showing both high and low virulence profiles. Monitoring the presence and distribution of MDV in these birds is essential, as they can serve as reservoirs for the virus and potentially trigger new outbreaks. Further research should be conducted to assess the virulence and pathogenicity of these strains in other hosts and bird breeds.

## Figures and Tables

**Figure 1 animals-14-02867-f001:**
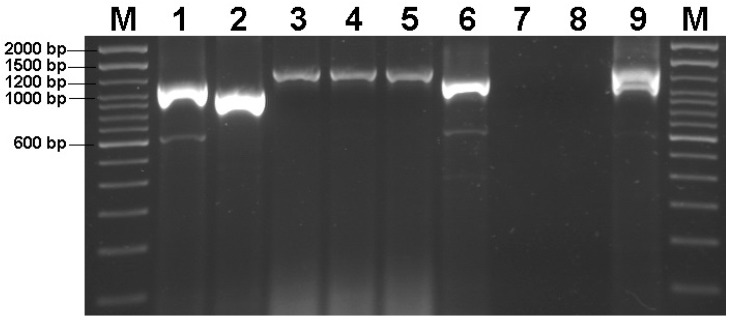
PCR amplification of the *meq* gene in the studied cases. Lanes: M (100 bp DNA Ladder, Invitrogen, Carlsbad, CA, USA); 1 (USP-386); 2 (USP-1171); 3 (USP-1540); 4 (USP-1790); 5 (USP-1873); 6 (USP-2429); 7 (USP-2583); 8 (negative control); 9 (positive control, CVI988 vaccine). In lines 1, 6, and 9, a PCR artifact of approximately 600 bp was observed.

**Figure 2 animals-14-02867-f002:**
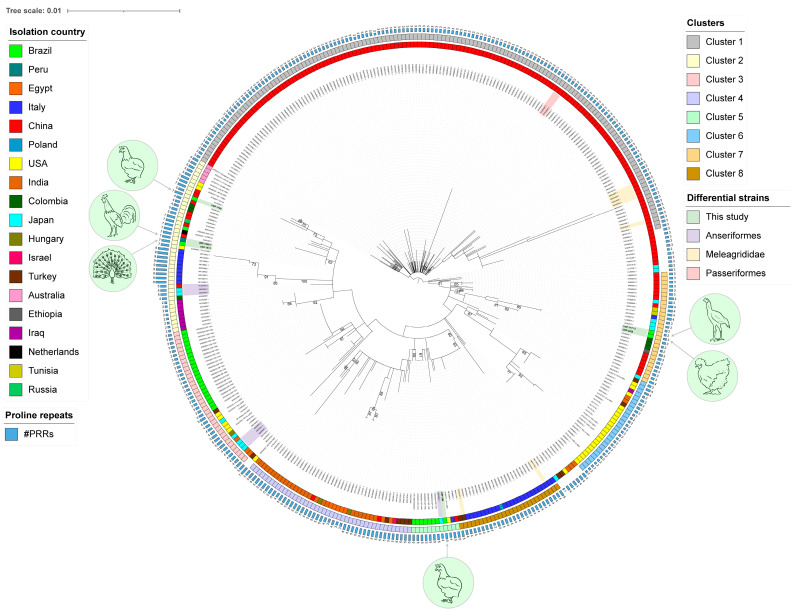
Maximum-likelihood phylogenetic tree of 387 complete *meq* gene sequences, including six strains from this study. The tree was inferred using a GTR + R substitution model with support values based on 1000 bootstrap replicates indicated on the branches. Additional annotations include phylogenetic cluster (inner ring), isolation country (middle ring), and number of PRRs (outer bars). The MDV strains from this study are highlighted in bold. The scale bar represents the number of substitutions per site.

**Table 2 animals-14-02867-t002:** Molecular detection of oncogenic viruses in the studied cases.

Case	ID	MDV	ALV	REV
1	USP-386	+	−	+
2	USP-1171	+	−	−
3	USP-1540	+	−	−
4	USP-1790	+	−	−
5	USP-1873	+	−	−
6	USP-2429	+	−	−
7	USP-2583	+	−	−

**Table 3 animals-14-02867-t003:** Molecular characteristics and amino acid substitutions of the *meq* protein of the studied strains compared with those of the MDV reference strains.

*meq* Domain ➔				BR			ZIP			TAD													
Amino Acid Site ^A^ ➔				71	77	80	110	115	119	139	151	153	176	180	182	194	216	217	277	283	320	326	337
Strain	Pathotype	Size ^B^ (aa)	PRRs N(N°)	A	E	Y	C	A	C	A	T	P	P	T	E	P	P	P	L	A	I	T	G
814	att	398	7	S	K	D	*	*	*	T	*	*	*	*	*	-	*	*	*	*	*	*	*
CVI988	att	398	7	S	*	D	*	V	*	T	*	*	*	*	*	-	*	*	*	*	*	I	*
CU-2	mMDV	398	7	S	*	D	*	V	*	T	*	*	*	*	*	-	*	*	*	*	*	*	*
BC-1	vMDV	398	7	S	A	D	*	*	*	T	*	*	*	*	*	-	*	*	*	*	*	*	*
FT158	vMDV	398	5	S	A	A	*	*	*	T	*	*	*	*	*	-	*	A	*	*	*	*	*
MPF57	vMDV	398	5	S	A	A	*	*	*	T	*	*	*	*	*	-	*	A	*	*	*	*	*
02LAR	vMDV	398	5	S	A	D	*	*	*	T	*	*	*	*	*	-	*	A	*	*	*	*	*
JM/102W	vMDV	399	7	*	*	D	*	V	*	T	*	*	*	*	*	*	S	*	*	*	*	*	*
04CRE	vMDV	398	5	S	A	D	*	*	*	T	*	*	*	*	*	-	*	A	*	*	*	*	*
GA	vMDV	339	5	*	K	D	*	V	*	T	*	*	*	*	*	*	*	*	*	*	*	*	*
571	vMDV	339	4	*	*	*	*	*	*	T	*	*	H	*	*	*	*	*	*	*	*	*	*
617A	vMDV	339	4	*	*	*	*	V	R	T	*	*	*	*	*	*	*	A	*	*	*	*	*
Woodlans1	vvMDV	398	5	S	A	A	*	*	*	T	*	*	*	*	*	-	*	A	*	*	*	*	*
Md5	vvMDV	339	4	*	K	D	*	V	*	T	*	*	*	*	*	*	*	A	*	V	T	*	*
549	vvMDV	339	2	*	K	D	*	V	R	T	*	Q	A	A	*	*	*	A	*	*	*	*	*
595	vvMDV	339	2	*	K	D	*	V	R	T	*	Q	A	A	*	*	*	A	*	*	*	*	*
643P	vvMDV	339	2	*	K	D	*	V	R	T	*	Q	A	A	*	*	*	A	F	*	*	*	*
RB-1B	vvMDV	339	5	*	K	D	*	V	*	T	*	*	*	*	*	*	*	*	*	*	*	*	*
W	vv+MDV	339	4	*	K	D	*	V	*	T	*	*	*	*	*	*	*	A	*	V	T	*	*
New	vv+MDV	339	2	*	K	D	*	V	R	T	*	Q	A	*	*	*	*	A	*	V	T	*	*
X	vv+MDV	339	2	*	K	D	*	V	R	T	*	Q	A	A	*	*	*	A	*	*	*	*	*
648A	vv+MDV	339	2	*	K	D	*	V	R	T	*	Q	A	A	*	*	*	A	P	*	*	*	*
USP-386		339	5	*	*	*	*	*	*	T	*	*	*	*	*	*	*	*	*	*	*	*	*
USP-1171		298	2	*	*	*	S	V	*	T	I	*	A	A	-	-	-	-	*	*	*	*	*
USP-1540		398	7	S	*	D	*	V	*	T	*	*	*	*	*	-	*	*	*	*	*	*	*
USP-1790		398	7	S	*	V	*	*	*	T	*	*	*	*	*	-	*	*	*	*	*	*	E
USP-1873		398	7	S	*	D	*	V	*	T	*	*	*	*	*	-	*	*	*	*	*	*	*
USP-2429		339	3	*	*	*	S	V	*	T	I	*	A	A	D	*	*	A	*	*	*	*	*
USP-1015 (G. I)		338	3	*	K	D	*	*	*	T	*	L	*	*	*	-	*	*	*	*	*	*	*
USP-1284 (G. II)		339	5	*	*	*	*	*	*	T	*	*	*	*	*	*	*	*	*	*	*	*	*
USP-1328 (G. III)		339	4	*	K	D	*	*	*	T	*	L	*	*	*	*	*	*	*	*	*	*	*
USP-1879 (G. IV)		398	7	S	*	D	*	V	*	T	*	*	*	*	*	-	*	*	*	*	*	I	*

^A^ Amino acid position according to the 339 aa-standard *meq* isoform of the strain Md5 (NC_002229). ^B^ The size of *meq* in amino acids does not account for the final codon (stop codon).

**Table 4 animals-14-02867-t004:** Sites under positive and negative selection pressure in the *meq* gene.

Positive Selection					Negative Selection			
Codon Position	FUBAR ^A^ Probability α < β	SLAC ^B^ P-[dN/dS > 1]	FEL ^B^ *p* Value	MEME ^B^ *p* Value	Codon Position	FUBAR ^A^ Probability α > β	SLAC ^B^ P-[dN/dS < 1]	FEL ^B^ *p* Value
3				0.00	49	0.999	0.00137	0.0013
71	0.966		0.0448	0.06	51			0.0311
77	0.968				55			0.0561
80	0.952				76			0.0741
88	0.987		0.0239	0.02	81			0.0932
101	0.905				106			0.0922
110	0.971		0.0827	0.07	114	0.923		0.0904
139	0.920				135			0.0419
151	0.907		0.0849		182	0.966	0.0343	0.0329
176	0.999	0.0177	0.0013	0.00	208	0.970	0.0370	0.0176
180	0.963		0.0589	0.08	225			0.0778
194	0.928		0.0724	0.09	253	0.975	0.0370	0.0080
203				0.03	269	0.995	0.0153	0.0031
205				0.00	298	0.997	0.00691	0.0009
217	0.995	0.0878	0.0118	0.00	333			0.0659
277				0.08				
285				0.01				
329				0.00				
339	0.927		0.0714	0.09				

^A^ FUBAR (*p* > 0.9). ^B^ MEME, FEL, SLAC (*p* < 0.1).

## Data Availability

The obtained sequences were submitted to GenBank under the accession numbers PP783759 to PP783764.

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
