# Peer review of "Diversity of Marek’s Disease Virus Strains in Infections in Backyard and Ornamental Birds"

_animals, 2024, doi:10.3390/ani14192867_

Round 1

Reviewer 1 Report

Comments and Suggestions for Authors

1. Results 3.1 should show the gross lesions and histopathologic examination in studied cases

2. Results 3.2 show molecular detection of oncogenic viruses

Figure 1 Lane 1 and Lane 8 two specific bands, why?

The strain USP-2583 could not be amplified by end-point PCR“ should be shown in Figure 1.

3.   Why didn't you isolate the virus? The sample USP-1873 is blood, which is suitable for virus isolation.

Author Response

Dear Editors and Reviewers:

We thank the reviewers for their thoughtful comments and constructive suggestions concerning our manuscript entitled “Diversity of Marek’s Disease Virus Strains in Infections in Backyard and Ornamental Birds” (ID: animals-3205474), which enabled us to resubmit an improved version of the manuscript. We highlighted the amendments in the revised manuscript, and responded, point by point to, the comments as listed below.

Reviewer #1:

Q1. Results 3.1 should show the gross lesions and histopathologic examination in studied cases

R1. We thank the reviewer for this observation. Due to the way the material was sent, unfortunately it was not possible to perform histopathological examinations. On the other hand, we have added more information specific to each case including findings of gross lesions and other details. These additions were added to section 2.1 as also requested by reviewer #3. The modifications are highlighted in yellow.

Q2. Results 3.2 show molecular detection of oncogenic viruses

R2. We thank the reviewer for this observation. As answered for the previous question, the information on gross lesions was included in section 2.1. In this way, it was not necessary to modify the numbering of the results for the molecular detection of oncogenic viruses (maintained as 3.1.).

Q3. Figure 1 Lane 1 and Lane 8 two specific bands, why?

R3. We thank the reviewer for this observation. In the case of line 1, it is a PCR artifact, especially in samples with high viral load. In the case of line 8, which corresponds to the CVI988 vaccine strain, the 2 fragments correspond to the standard and long isoforms of meq. This has already been reported in this strain and is associated with the passage and attenuation process. We have added this information in section 3.1. The modifications are highlighted in yellow.

Q4. “The strain USP-2583 could not be amplified by end-point PCR“ should be shown in Figure 1.

R4. We thank the reviewer for this observation. We have now included sample USP-2583 in Figure 1 as suggested. The modifications are highlighted in yellow.

Q5.  Why didn't you isolate the virus? The sample USP-1873 is blood, which is suitable for virus isolation.

R5. We thank the reviewer for this observation. The blood sample as well as the sample from Peru were received impregnated in FTA Whatman Cards and it was not possible to use them for viral isolation. We have now detailed this information in table 1. The modifications are highlighted in yellow.

Reviewer 2 Report

Comments and Suggestions for Authors

Line 67- please remove Nair et al. as they are referenced appropriately.

Please consider adding a section on vaccination.

Line 83- Please consider the abbreviation “Marek's disease” to stay consistent throughout the manuscript.

Line 101- Please consider removing GE Healthcare and revise accordingly- an Illustra™ _GFX PCR and Gel Band Purification Kit (GE Healthcare).

Line 105- Please specify the software used for translating the RNA transcript into the appropriate amino acid sequence.

Figure 1- Please specify where the DNA ladder was purchased from. The labeling in the ladder bands is slightly off, given the description in the text (lines 131-137). The brightest band in a 100 bp ladder is typically at 500 bp. Per the manuscript, lanes 1 and 6 should correspond to a band size of 1020bp. Figure 1 indicates a larger size. Please ensure that the bands are labeled correctly.

Additionally, lane 7 should have been loaded by sample USP- 2583 and lanes 8 and 9 for the controls.

Please consider explaining the expected fragment sizes for the positive control CVI988 vaccine.

Line 136- The strain USP-2583 could not be amplified by end-point PCR. Please consider discussing the reason behind this result/outcome.

Line 153- USP-1171 (due to the deletion of 42 amino acids). Please consider expanding this for clarity and accuracy as 42 amino acid X 3 = 126bp and 126bp+897bp= 1023bp.

In Table 3 USP1171 is at 298 aa i.e. 298X3= 894bp

USP-1171

298

Comments on the Quality of English Language

Line 75-79: Please consider rewriting the last paragraph for clarity.

Line 99- Please consider adding a comma after strains: To characterize the detected MDV strains, the meq gene was amplified according to 99 previously reported PCR conditions [24].

Author Response

Dear Editors and Reviewers:

We thank the reviewers for their thoughtful comments and constructive suggestions concerning our manuscript entitled “Diversity of Marek’s Disease Virus Strains in Infections in Backyard and Ornamental Birds” (ID: animals-3205474), which enabled us to resubmit an improved version of the manuscript. We highlighted the amendments in the revised manuscript, and responded, point by point to, the comments as listed below.

Reviewer #2:

Q1. Line 67- please remove Nair et al. as they are referenced appropriately.

R1. We thank the reviewer for this observation. We have corrected this typing error. The modification is highlighted in yellow.

Q2. Please consider adding a section on vaccination.

R2. We thank the reviewer for this observation. We have now added a section on vaccination in MDV as suggested. The modifications are highlighted in yellow.

Q3. Line 83- Please consider the abbreviation “Marek's disease” to stay consistent throughout the manuscript.

R3. We thank the reviewer for this observation. We have now changed to the abbreviation MD as suggested. The modification is highlighted in yellow.

Q4. Line 101- Please consider removing GE Healthcare and revise accordingly- an Illustra™ _GFX PCR and Gel Band Purification Kit (GE Healthcare).

R4. We thank the reviewer for this observation. We have corrected and updated the kit information. The modifications are highlighted in yellow.

Q5. Line 105- Please specify the software used for translating the RNA transcript into the appropriate amino acid sequence.

R5. We thank the reviewer for this observation. We have updated the software information and procedure as suggested. The modifications are highlighted in yellow.

Q6. Figure 1- Please specify where the DNA ladder was purchased from. The labeling in the ladder bands is slightly off, given the description in the text (lines 131-137). The brightest band in a 100 bp ladder is typically at 500 bp. Per the manuscript, lanes 1 and 6 should correspond to a band size of 1020bp. Figure 1 indicates a larger size. Please ensure that the bands are labeled correctly.

R6. We thank the reviewer for this observation. We have added information about the DNA ladder in the legend of Figure 1 as suggested. The most intense band corresponds to 600 bp, as can be seen on the product site: https://www.thermofisher.com/order/catalog/product/15628019. The PCR used amplifies the complete meq gene plus 128 bp on the flanks, due to this the sizes observed in the electrophoresis are larger. We have added this clarification in section 2.2. of the methods and in 3.1. of the results. The modifications are highlighted in yellow.

Q7. Additionally, lane 7 should have been loaded by sample USP- 2583 and lanes 8 and 9 for the controls.

R7. We thank the reviewer for this observation. We have updated the figure including sample USP-2583 and lanes for controls as suggested. The modifications are highlighted in yellow.

Q8. Please consider explaining the expected fragment sizes for the positive control CVI988 vaccine.

R8. We thank the reviewer for this observation. The amplification of two meq isoforms in the vaccine strain CVI988 has been reported several times. We have added clarification and details of this result in section 3.1. The modifications are highlighted in yellow.

Q9. Line 136- The strain USP-2583 could not be amplified by end-point PCR. Please consider discussing the reason behind this result/outcome.

R9. We thank the reviewer for this observation. We consider that non-amplification is due to the low viral load in the sample. We have added a sentence to justify this result in section 3.1. as suggested. The modifications are highlighted in yellow.

Q10. Line 153- USP-1171 (due to the deletion of 42 amino acids). Please consider expanding this for clarity and accuracy as 42 amino acid X 3 = 126bp and 126bp+897bp= 1023bp.

R10. We thank the reviewer for this observation. The size of the deleted region is 41 amino acids (1020-897=123bp=41aa) as well noted. We have corrected the value as suggested. We apologize for this error. The modifications are highlighted in yellow.

Q11. In Table 3 USP1171 is at 298 aa i.e. 298X3= 894bp. USP-1171 - 298

R11. We thank the reviewer for this observation. The size of meq in amino acids does not account for the final codon (stop codon) since it does not produce an amino acid. We have added this clarification in the legend of Table 3. The modifications are highlighted in yellow.

Q12. Line 75-79: Please consider rewriting the last paragraph for clarity.

R12. We thank the reviewer for this observation. We have modified these lines as suggested. The modifications are highlighted in yellow.

Q13. Line 99- Please consider adding a comma after strains: To characterize the detected MDV strains, the meq gene was amplified according to 99 previously reported PCR conditions [24].

R13. We thank the reviewer for this observation. We have added the comma as suggested. The modification is highlighted in yellow.

Reviewer 3 Report

Comments and Suggestions for Authors

Peer review: 

Manuscript "Diversity of Marek’s Disease Virus Strains in Infections in Backyard and Ornamental Birds” (animals-3205474)

Comments to the authors:

The study aimed to report the detection of Marek’s Disease Virus (MDV) in backyard and ornamental birds from Brazil and Peru as well as to carry out a molecular characterization of the meq oncogene of the detected strains. According to the authors, Marek disease is caused by Mardivirus gallidalpha2. It infects various bird species resulting in diverse clinical manifestations. Viral meq gene is necessary for oncogenesis and it was sequenced in the study. The results demonstrated the detection of MDV in seven outbreaks. Three isoforms of meq (S-meq, meq, and L-meq) and 2-7 proline repeat regions (PRRs) were detected in the sequenced strains. At the amino acid level, profiles with low and high virulence potential were identified. Phylogenetic analysis classified the viral strains into three distinct clusters, highlighting MDV diversity in the studied birds. 

In my opinion, the study was well designed and has interesting results. The methods are relatively well described and the results are presented in interesting Tables/Figures. However, the manuscript is not ready and requires several improvements. I am highlighting some main necessary modifications:

1) Introduction: the epidemiological scenario is very poorly explained! It is presented in two very concise sentences (lines 75-76). It is necessary to better describe the whole epidemiological situation in South America. There are at least 6 references (the authors cited them). Please expand this paragraph! 

2) Sampling population: the clinical cases are not clearly described. If there were outbreaks, (“The present study included a collection of cases of outbreaks involving backyard and ornamental birds affected by Marek's disease”), please explain them in more detail! Where? How many birds? What were the clinical consequences for the birds? 

3) Methods: the description of the methods is very concise, citing other previous studies. I would suggest to expand the description of the main methods for a better understanding of the study. It is also mandatory to deposit the obtained meq sequences in the Genbank and to report them in the manuscript. It is also recommended to provide the dataset used for the phylogenetic analyses as supplemental material. 

4) Results: Figure 2 is very interesting. However, the complete description of the results could be improved and better ordered. I suggest to present first the phylogenetic results and after the main amino acid indels / substitutions. It is more logical. 

Finally, the Discussion should be revised after all modifications. After the preparation of a new version, the manuscript needs to be peer reviewed again. .  

Comments on the Quality of English Language

Moderate editing of English language required.

Author Response

Dear Editors and Reviewers:

We thank the reviewers for their thoughtful comments and constructive suggestions concerning our manuscript entitled “Diversity of Marek’s Disease Virus Strains in Infections in Backyard and Ornamental Birds” (ID: animals-3205474), which enabled us to resubmit an improved version of the manuscript. We highlighted the amendments in the revised manuscript, and responded, point by point to, the comments as listed below.

Reviewer #3:

Q0. The study aimed to report the detection of Marek’s Disease Virus (MDV) in backyard and ornamental birds from Brazil and Peru as well as to carry out a molecular characterization of the meq oncogene of the detected strains. According to the authors, Marek disease is caused by Mardivirus gallidalpha2. It infects various bird species resulting in diverse clinical manifestations. Viral meq gene is necessary for oncogenesis and it was sequenced in the study. The results demonstrated the detection of MDV in seven outbreaks. Three isoforms of meq (S-meq, meq, and L-meq) and 2-7 proline repeat regions (PRRs) were detected in the sequenced strains. At the amino acid level, profiles with low and high virulence potential were identified. Phylogenetic analysis classified the viral strains into three distinct clusters, highlighting MDV diversity in the studied birds.

In my opinion, the study was well designed and has interesting results. The methods are relatively well described and the results are presented in interesting Tables/Figures. However, the manuscript is not ready and requires several improvements. I am highlighting some main necessary modifications:

R0. We thank the reviewer for thoughtful comments and constructive suggestions concerning our manuscript.

Q1. Introduction: the epidemiological scenario is very poorly explained! It is presented in two very concise sentences (lines 75-76). It is necessary to better describe the whole epidemiological situation in South America. There are at least 6 references (the authors cited them). Please expand this paragraph!

R1. We thank the reviewer for this observation. We have expanded this paragraph as suggested. The modifications are highlighted in yellow.

Q2. Sampling population: the clinical cases are not clearly described. If there were outbreaks, (“The present study included a collection of cases of outbreaks involving backyard and ornamental birds affected by Marek's disease”), please explain them in more detail! Where? How many birds? What were the clinical consequences for the birds?

R2. We thank the reviewer for this observation. We have added more information specific to each case including findings of gross lesions and other details. These additions were added to section 2.1. The modifications are highlighted in yellow.

Q3. Methods: the description of the methods is very concise, citing other previous studies. I would suggest to expand the description of the main methods for a better understanding of the study. It is also mandatory to deposit the obtained meq sequences in the Genbank and to report them in the manuscript. It is also recommended to provide the dataset used for the phylogenetic analyses as supplemental material.

R3. We thank the reviewer for this observation. We have now extended the descriptions in each method section as suggested. We have also added a sentence with the GenBank codes of the sequences. Also, we have included the list of sequences used with accession codes and other information as supplementary material (Table S1). The modifications are highlighted in yellow.

Q4. Results: Figure 2 is very interesting. However, the complete description of the results could be improved and better ordered. I suggest to present first the phylogenetic results and after the main amino acid indels / substitutions. It is more logical.

R4. We thank the reviewer for this observation. We are now presenting first the results of the phylogenetic analysis and then the amino acid polymorphisms as suggested. The modifications are highlighted in yellow.

Q5. Finally, the Discussion should be revised after all modifications. After the preparation of a new version, the manuscript needs to be peer reviewed again.

R5. We thank the reviewer for this observation. We have modified and extended some sentences in the discussions according to additional information about the cases. The modifications are highlighted in yellow.

Round 2

Reviewer 3 Report

Comments and Suggestions for Authors

I have already reviewed the first submission of this manuscript. The study reports the detection of Marek's disease virus (MDV) in ornamental and backyard birds from Brazil and Peru. In addition, the meq oncogene was sequenced to characterize the strains. The results demonstrated the detection of MDV in seven outbreaks. Three meq isoforms (S-meq, meq and L-meq) and 2-7 proline repeat regions (PRRs) were detected in the sequenced strains. At the amino acid level, profiles with low and high virulence potential were identified. Phylogenetic analysis classified the viral strains into three distinct groups, highlighting the diversity of MDV in the ornamental and backyard birds studied.

As I mentioned, the study presents interesting data. The methods are relatively well described and the results are presented in well-prepared Tables/Figures. The authors also addressed some important concerns that I mentioned in my first review as follows:

1. Introduction: the epidemiological scenario was better explained.

BUT: The text is still not very well organized sequentially and further improvements are needed. Two examples: (a) clinical manifestations are mixed with control in the same paragraph (the 4th), (b) epidemiology is described in two paragraphs very far apart from each other (the second and fifth) that would be much better together. Therefore, I suggest an additional revision to improve the Introduction, focusing on a more logical and well-ordered presentation of the information.

2. Sample population: Clinical cases are better explained. But the authors need to be careful not to duplicate information in the text and in Table 1. Please review the text and the Table 1 avoiding repetition of the information.

3. Methods: The description of the methods has been improved. The authors have also deposited and reported the meq sequences in Genbank.

4. Results: They are more logically ordered now.

BUT: The explanations of the photo (lines 210-212) should be in the figure statement. And not in the text of the Results section. Additionally, the final sentence of this paragraph (lines 213-214) should be moved to the Discussion section.

Finally, the English is difficult to understand in some sentences. There are also many sentences with redundancy and repetitions of the same words. Therefore, additional proofreading of the English language is necessary throughout the text.

Comments on the Quality of English Language

The English is difficult to understand in some sentences. There are also many sentences with redundancy and repetitions of the same words. Therefore, additional proofreading of the English language is necessary throughout the text.

Author Response

Dear Editors and Reviewers:

We thank the reviewers for their thoughtful comments and constructive suggestions concerning our manuscript entitled “Diversity of Marek’s Disease Virus Strains in Infections in Backyard and Ornamental Birds” (ID: animals-3205474), which enabled us to resubmit an improved version of the manuscript. We highlighted the amendments in the revised manuscript, and responded, point by point to, the comments as listed below.

Reviewer #1:

Q0. I have already reviewed the first submission of this manuscript. The study reports the detection of Marek's disease virus (MDV) in ornamental and backyard birds from Brazil and Peru. In addition, the meq oncogene was sequenced to characterize the strains. The results demonstrated the detection of MDV in seven outbreaks. Three meq isoforms (S-meq, meq and L-meq) and 2-7 proline repeat regions (PRRs) were detected in the sequenced strains. At the amino acid level, profiles with low and high virulence potential were identified. Phylogenetic analysis classified the viral strains into three distinct groups, highlighting the diversity of MDV in the ornamental and backyard birds studied.

As I mentioned, the study presents interesting data. The methods are relatively well described and the results are presented in well-prepared Tables/Figures. The authors also addressed some important concerns that I mentioned in my first review as follows.

R0. We thank the reviewer for thoughtful comments and constructive suggestions concerning our manuscript.

Q1. Introduction: the epidemiological scenario was better explained.

BUT: The text is still not very well organized sequentially and further improvements are needed. Two examples: (a) clinical manifestations are mixed with control in the same paragraph (the 4th), (b) epidemiology is described in two paragraphs very far apart from each other (the second and fifth) that would be much better together. Therefore, I suggest an additional revision to improve the Introduction, focusing on a more logical and well-ordered presentation of the information.

R1. We thank the reviewer for this observation. We have rearranged the order of the introduction as suggested. The modifications are highlighted in yellow.

Q2. Sample population: Clinical cases are better explained. But the authors need to be careful not to duplicate information in the text and in Table 1. Please review the text and the Table 1 avoiding repetition of the information.

R2. We thank the reviewer for this observation. We have modified the text and table 1 to minimize repetition of information as suggested. The modifications are highlighted in yellow.

Q3. Methods: The description of the methods has been improved. The authors have also deposited and reported the meq sequences in Genbank.

R3. We thank the reviewer for thoughtful comments and constructive suggestions concerning our manuscript.

Q4. Results: They are more logically ordered now.

BUT: The explanations of the photo (lines 210-212) should be in the figure statement. And not in the text of the Results section. Additionally, the final sentence of this paragraph (lines 213-214) should be moved to the Discussion section.

R4. We thank the reviewer for this observation. We have transferred the information indicated in the figure legend and the last sentence of that paragraph to the discussions as suggested. The modifications are highlighted in yellow.

Q5. Finally, the English is difficult to understand in some sentences. There are also many sentences with redundancy and repetitions of the same words. Therefore, additional proofreading of the English language is necessary throughout the text.

Comments on the Quality of English Language

The English is difficult to understand in some sentences. There are also many sentences with redundancy and repetitions of the same words. Therefore, additional proofreading of the English language is necessary throughout the text.

R5. We thank the reviewer for this observation. We have revised the language throughout the manuscript as suggested. The modifications are highlighted in yellow.